# Effect of Ambient Air Pollution on Hospital Readmissions among the Pediatric Asthma Patient Population in South Texas: A Case-Crossover Study

**DOI:** 10.3390/ijerph17134846

**Published:** 2020-07-06

**Authors:** Juha Baek, Bita A. Kash, Xiaohui Xu, Mark Benden, Jon Roberts, Genny Carrillo

**Affiliations:** 1Department of Environmental and Occupational Health, School of Public Health, Texas A&M University, College Station, TX 77843, USA; jbaek@tamu.edu (J.B.); mthowdy@tamu.edu (M.B.); 2Center for Outcomes Research, Houston Methodist Research Institute, Houston, TX 77030, USA; bakash@tamu.edu; 3Center for Health & Nature, Houston Methodist Research Institute, Houston, TX 77030, USA; 4Department of Health Policy and Management, School of Public Health, Texas A&M University, College Station, TX 77843, USA; 5Department of Epidemiology and Biostatistics, School of Public Health, Texas A&M University, College Station, TX 77843, USA; xiaohui.xu@tamu.edu; 6Department of Pediatric Pulmonology, Driscoll Children’s Hospital, Corpus Christi, TX 78411, USA; Jon.Roberts@dchstx.org

**Keywords:** hospital readmissions, pediatric asthma, ambient air pollution, PM_2.5_, ozone, South Texas, low-income communities

## Abstract

Few studies have evaluated the association between ambient air pollution and hospital readmissions among children with asthma, especially in low-income communities. This study examined the short-term effects of ambient air pollutants on hospital readmissions for pediatric asthma in South Texas. A time-stratified case-crossover study was conducted using the hospitalization data from a children’s hospital and the air pollution data, including particulate matter 2.5 (PM_2.5_) and ozone concentrations, from the Centers for Disease Control and Prevention between 2010 and 2014. A conditional logistic regression analysis was performed to investigate the association between ambient air pollution and hospital readmissions, controlling for outdoor temperature. We identified 111 pediatric asthma patients readmitted to the hospital between 2010 and 2014. The single-pollutant models showed that PM_2.5_ concentration had a significant positive effect on risk for hospital readmissions (OR = 1.082, 95% CI = 1.008–1.162, *p* = 0.030). In the two-pollutant models, the increased risk of pediatric readmissions for asthma was significantly associated with both elevated ozone (OR = 1.023, 95% CI = 1.001–1.045, *p* = 0.042) and PM_2.5_ concentrations (OR = 1.080, 95% CI = 1.005–1.161, *p* = 0.036). The effects of ambient air pollutants on hospital readmissions varied by age and season. Our findings suggest that short-term (4 days) exposure to air pollutants might increase the risk of preventable hospital readmissions for pediatric asthma patients.

## 1. Introduction

Hospital readmission is an important asthma-related health outcome to consider, since repeated hospitalizations could lead to a large burden on patients/caregivers, hospitals, and the government with regard to health care resources, cost and quality [1,2]. In addition, a study found that children who are rehospitalized for asthma might differ in disease severity, access to care, or environmental exposures when compared with those who were hospitalized only once [3]. However, asthma readmissions can be potentially prevented with appropriate and timely primary care given that asthma is regarded as one of the ambulatory care-sensitive conditions [4,5,6]. As such, reducing preventable hospital readmissions is a key priority in order to decrease health care costs and improve quality of care and patient experience [7].

Exposure to air pollution has been regarded as a significant trigger for asthma exacerbation that may lead to repeated hospitalizations. Studies have classified air pollution into two types: indoor and outdoor. With regard to indoor air pollution, common air pollutants include moisture, smoking, dust, and chemicals—these can all serve as asthma triggers in children. Several studies found that exposure to indoor environmental conditions, such as dust mites, moisture and mold, was a significant factor for pediatric asthma exacerbations in the home setting [8,9].

Further, two studies specifically examined the relationship between indoor air quality and hospital readmissions for children with asthma. One study showed that exposure to tobacco in the indoor setting, such as detectable serum or salivary cotinine collected by biomarkers, was revealed to be significantly associated with increased risk of hospital readmission [10]. The second study found that detection of poor air quality, such as higher levels of fungi and yeast, in a child’s bedroom increased the risk of hospital readmission [11]. The study also reported that having a carpeted floor in the bedroom as well as a high frequency of vacuuming at home were both significantly related with increased chance of rehospitalization for children with asthma [11].

In the case of outdoor air pollution, this subject has been studied and considered as a significant factor leading to hospital readmission among children with asthma. Previous studies have shown that exposure to traffic-related air pollution (TRAP) or residential proximity to major roads was associated with adverse respiratory health effects in children, including exacerbation of asthma symptoms [12,13,14]. Other studies have also examined the relationship between TRAP exposure and hospital readmission for children with asthma or bronchodilator-responsive wheezing [15,16,17]. In particular, one study found that higher TRAP exposure was significantly associated with greater risk of readmission in the unadjusted model, but this relationship was not significant in the adjusted model [15].

However, there is a paucity of studies that evaluate the impacts of short-term exposure to specific ambient air pollutants other than TRAP on preventable hospital readmissions among pediatric asthma patients. This lack of understanding is especially true for relatively low-income communities. To address this gap in the literature, this study examined the effects of ambient air pollutants, including PM_2.5_ and ozone concentrations, on the risk of preventable hospital readmission for pediatric asthma patients in South Texas.

## 2. Materials and Methods

### 2.1. Data Sources and Study Setting

Hospitalization data for pediatric asthma between 1 January 2010 and 31 December 2014 were collected from Driscoll Children’s Hospital database. The hospital, which focuses on pediatric care, is located in the city of Corpus Christi, and has 189 beds serving the children of South Texas. The information recorded on the data included basic demographics (gender, age, ethnicity), type of insurance, dates of admission and discharge, the International Classification of Diseases 9th Revision (ICD-9) diagnosis code, and census tract information of each patient’s residence. The inclusion criteria of the study participants were children aged 5 to 18 years old and those who were readmitted to the hospital due to asthma as a primary diagnosis (ICD-9, 493) during the study period. The subsequent admissions after index hospitalization for each patient were included as readmission cases in this study. This study protocol was reviewed and approved by the Institutional Review Boards of the Texas A&M University (IRB2018-0857D) and Driscoll Children’s Hospital.

Data on ambient air pollutants during the study period were gathered from the Centers for Disease Control and Prevention (CDC) environmental public health tracking network [18]. The daily average predicted atmospheric particulate matter 2.5 (or PM_2.5_) and ozone concentrations in the census tract level were estimated from the Downscaler model of the United States Environmental Protection Agency (EPA) [19]. The data used in this study were the average daily air pollution concentration, including PM_2.5_ and ozone, given the data availability. The data for PM_2.5_ and ozone indicate the mean estimated 24 hour average concentration (µg/m^3^) and the mean estimated 8 hour average concentration within three meters of the surface of the earth (parts per billion [ppb]), respectively [20]. The meteorological data of daily mean temperatures were collected from the Texas Commission on Environmental Quality (TCEQ) in order to adjust for the impact of temperature on the association between ambient air pollution and pediatric asthma rehospitalization. The geographic information system (GIS) program (ArcMap 10.4, ESRI, Redlands, CA) was used to obtain the temperature data measured in the nearest air monitoring station from each patient’s residence. 

### 2.2. Study Design and Measurement

A time-stratified case-crossover study design was applied to evaluate the short-term effects of ambient air pollution on hospital readmission for children with asthma. In this study, air pollution data for each patient were collected for the case period, which indicates the week of the readmission date, and three control periods, which refer to the week before, the week after, and two weeks after the readmission date. Namely, each of two air pollutant concentrations (PM_2.5_ and ozone) was collected from readmission day (Lag0) to three days before readmission (Lag1–Lag3) in the week of hospital readmission for the case period and the corresponding days (4 days: Lag0–Lag3) in each of the three weeks for control periods. 

To assess the relationships between ambient air pollutants and hospital readmission, we fit the models with different lag structures between the readmission day (Lag0) and three single-day lags (Lag1–Lag3). We also evaluated the associations with cumulative-day lags, indicating 2 day (Lag0–1), 3 day (Lag0–2), and 4 day (Lag0–3) moving averages of PM_2.5_ and ozone concentrations, since single-day lag models might underestimate the relationships [21]. For example, Lag0–2 would refer to the averaged values of each air pollutant in a total of 3 days from Lag0 to Lag2 for the case period and control periods. The air pollution exposure during the case period was compared with that of the control periods for each participant. 

In addition to single-pollutant models, two-pollutant models were conducted to investigate the effect of each air pollutant on hospital readmission, controlling for the other air pollutant. Potential confounders such as individual-level characteristics were controlled by the study design, since case and control periods are compared for the same patient [22]. However, temperature information was included as a time-varying factor in the analysis. 

### 2.3. Statistical Analysis

Demographic characteristics of the study population were calculated to estimate the mean and standard deviation (SD) for continuous variables and percentages for categorical variables. Pearson correlation tests were performed to assess whether ambient air pollutants (PM_2.5_ and ozone) and temperature are highly correlated. A conditional logistic regression analysis was conducted to examine associations between ambient air pollution and the odds of a hospital readmission for pediatric asthma. We controlled temperature for the same periods as air pollution in all of the models. The results are presented as adjusted odds ratios (ORs) and 95% confidence intervals (CIs). Stratified analyses by age (5–11 years old or 12–18 years old), gender (girls or boys) and season (warm: May–October or cold: November–April) were used to assess effects as a modifier. All analyses were conducted by using the Stata 14 version (StataCorp LLC, College Station, TX). All statistical tests were two sided, and a *p*-value < 0.05 was considered to be statistically significant. 

## 3. Results

A total of 111 patients were readmitted to the children’s hospital due to asthma between 2010 and 2014 (Table 1). The average age was approximately 9 years old and the number of males was higher than females (57.6 vs. 42.3%). Most of the patients readmitted to the hospital were Hispanic (79.3%) and approximately 70% had public insurance (Medicaid). The average time to readmission was approximately 386 days, only 8.1% of patients had 30 day readmission and approximately 37% were readmitted to the hospital in 1 year or longer. The readmissions in the cold season were a little higher than those in the warm season (52.3% vs. 47.7%).

Table 2 displays summary statistics of daily ambient air pollutant concentrations and daily temperature in South Texas between 2010 and 2014. The overall mean of PM_2.5_ and ozone concentrations was 8.3 (μg/m^3^) and 37.37 (ppb), respectively. The overall average temperature in the study region was 20.2 ℃ and its range was between 8.83 and 29.73 ℃. The results of the Pearson correlation tests showed that ambient air pollutant and temperature were not highly correlated with each other (correlation coefficient r = −0.214 to 0.077, *p* < 0.05) (See Appendix A
Table A1). 

Table 3 describes the results of the conditional logistic regression analysis to examine the association between ambient air pollution and preventable hospital readmission. We found that the elevated PM_2.5_ concentration was significantly associated with an increased risk of preventable hospital readmission on Lag1 in both single-pollutant (OR = 1.082, 95% CI = 1.008–1.162, *p* = 0.030) and two-pollutant models (OR = 1.080, 95% CI = 1.005–1.161, *p* = 0.036), controlling for temperature. Further, we observed a significant positive association of preventable hospital readmission with ozone concentration on Lag0 in the two-pollutant model (OR = 1.023, 95% CI = 1.001–1.045, *p* = 0.042). However, none of the ambient air pollutants were significant on cumulative-day lags (Lag0–1 ~ Lag0–3) in the single- and two-pollutant models. Figure 1 illustrates the lag structures of ORs and 95% CIs of PM_2.5_ and ozone concentrations with preventable hospital readmission for the patients on single- and multiple-day lags in the single- and two-pollutant models. 

Table 4 demonstrates the results of age-stratified conditional logistic regression models. We found that the association between ozone concentration and risk of preventable readmission was significantly positive on Lag0 in the two-pollutant model among children aged 5–11 years, adjusting for temperature (OR = 1.029, 95% CI = 1.004–1.055, *p* = 0.022). Yet, none of the associations for PM_2.5_ concentration were significant. On the other hand, there was no significant relationship of ambient air pollutants with preventable hospital readmission among those aged 12–18 years old. The gender-stratified models shown in Table 5 show that associations between ambient air pollutants and pediatric asthma rehospitalization were not significant among males and females.

Table 6 displays the results of the conditional logistic regression analysis stratified by season. We observed that both PM_2.5_ and ozone had more of an effect on the risk of preventable rehospitalization during the warm season (May–October) compared with the cold season (November–April). Specifically, the association between PM_2.5_ and preventable readmission was found to be positively significant on Lag1 in the single-pollutant (OR = 1.134, 95% CI = 1.019–1.262, *p* = 0.021) and two-pollutant models (OR = 1.125, 95% CI = 1.009–1.254, *p* = 0.034) as well as on Lag0–1 in the single-pollutant model (OR = 1.146, 95% CI = 1.003–1.309, *p* = 0.046) in the warm season. For ozone, positive relationships with hospital readmission were observed on Lag0 in both single-pollutant (OR = 1.043, 95% CI = 1.012–1.075, *p* = 0.007) and two-pollutant models (OR = 1.043, 95% CI = 1.010–1.078, *p* = 0.01), and on Lag0–1 in the single-pollutant model (OR = 1.033, 95% CI = 1.002–1.065, *p* = 0.037) in the warm season. No significant associations were found in the cold season. 

## 4. Discussion

This study examined the effects of ambient air pollution on preventable hospital readmission among children with asthma in South Texas. We found that short-term exposure to PM_2.5_ concentration was positively associated with an increased risk of preventable rehospitalization for pediatric asthma in both the single- and two-pollutant models. Ozone pollutant also had a significantly positive effect on asthma hospital readmission in the two-pollutant model. Specifically, significant associations between ambient air pollutants and hospital readmissions were found in the readmission day (Lag0) and the day before readmission day (Lag1). 

The findings of this study provide support for the adverse effects of ambient air pollutants on the risk of preventable hospital readmission for pediatric asthma. This is important new knowledge, given that previous studies only included traffic-related air pollutants (TRAPs). For example, a study reported that residential exposure to specific TRAPs including NO_x_ and CO was significantly associated with an elevated risk of rehospitalizations for pediatric asthma [16]. Another study targeting children with asthma or bronchodilator-responsive wheezing showed that exposure to TRAPs affected hospital readmission, especially among white children [15]. In addition, one other study showed that the residential proximity to major roads or freeways (within 300 meters) increased the risk of hospital readmission for pediatric asthma [17]. As a result, the current study extends the findings of earlier evidence about a relationship between outdoor air pollutants and hospital readmissions for asthma.

The results of this study were consistent with those of previous studies that explored the effect of ambient air pollutants on pediatric asthma exacerbations, including hospital admission and emergency department (ED) visits. A review of 22 studies reported that exposures to PM_2.5_, sulfur dioxide (SO_2_), and nitrogen dioxide (NO_2_) were significantly associated with asthma exacerbation among children less than 18 years of age [23]. Multiple studies have also found positive relationships between PM_2.5_ and asthma hospitalization or ED visits for pediatric patients [24,25,26,27]. Moreover, previous research revealed the adverse effects of ozone concentration on asthma exacerbations for children in different settings, such as Texas [28], New York [29], California [30], China [24], HongKong [31] and Korea [32]. 

Based on this study, the effects of ambient air pollutants on preventable hospital readmission varied by age and season. We found that ozone concentration may contribute to increasing the odds of preventable hospital readmission for younger children aged 5–11 years but not for those aged 12–18 years. There is no evidence that substantiates the effects of PM_2.5_ or ozone on pediatric asthma rehospitalization by age; however, one study presented that traffic-related NO_x_ and CO had a significant age effect on readmission for infants but not for children aged 1 to 18 years [16]. Age-stratified models of studies reporting the impact of PM_2.5_ and ozone on asthma pediatric exacerbations had mixed results in different age groups. Studies found a significant association between air pollutants (PM_2.5_ and/or ozone) and asthma exacerbations among preschool children (5 years or less) [25,26,33], those aged 5–14 years [25,34], and those aged 6–18 years [24,33]. 

The models stratified by season showed significant positive effects of PM_2.5_ and ozone on preventable hospital readmission during the warm season, while such effects were not observed during the cold season. Our results were consistent with those of previous studies that examined the association between ambient air pollutants and asthma exacerbations for pediatric asthma. Most studies that were previously conducted found a significant effect of PM_2.5_ on asthma hospitalization or ED visits during the warm season (May–October or April–September), and none in the cold season [25,26,35,36]. Another study also found a stronger association in the warm season than in the cold season [34]. Additionally, other studies have also revealed a significant relationship between ozone and asthma exacerbations in the warm season [35,37,38]. This finding is most likely due to the fact that children tend to play outside or have more outdoor activities at home and at school during the warm season compared to the cold season. Their houses are also more likely to be ventilated during the warm season, which may increase their personal exposure to ambient air pollutants [25,39,40]. 

The models stratified by gender did not show any significant associations in the current study. This finding is contrary to previous research, which suggested that gender could be a modifying factor of the relationship between TRAPs and hospital readmission. One study found that TRAPs including NO_x_ and CO were associated with pediatric asthma rehospitalizations among females only [16]. Another study revealed that the adverse impact of residence near heavy traffic on repeated asthma hospitalization was stronger among females than males (in the 6–18 years of age group) [17]. The gender difference was also observed in previous studies that investigated the relationship between ambient air pollutants (PM_2.5_ and ozone) and asthma exacerbations for pediatric asthma [24,25,26,33].

### 4.1. Limitations

This study has several limitations. First, we used daily average air pollutant concentrations predicted by the statistical model at the census tract level as a proxy for personal exposure to outdoor air pollution, which may cause an exposure measurement error and thus underestimate the impacts of ambient air pollutants [41]. Second, the temperature data may not be accurate, since the distance from the residence to the closest monitoring sites varied for each patient. This may also lead to measurement error. Third, only one children’s hospital in South Texas, where ambient air pollutant concentrations were relatively low, was included in this study. As such, the findings may not be generalizable to other settings in different regions. Further study should be conducted in other regions with higher ambient air pollution to confirm the association. Fourth, some important factors that may potentially be associated with personal exposure to air pollution, such as the amount of time spent for outdoor activities and the level of indoor air pollution, and other air pollutants like NO_2_ were not included in this study due to unavailability of data. Finally, air pollution data were linked to the patients’ residential census tract; however, we cannot guarantee that the patients were actually living or staying in that address during the study period. 

### 4.2. Health Policy and Practice Implications

The findings of this study have several important implications for public health and health care services to reduce preventable hospital readmission. First, it would be important for children with asthma, who have experienced hospitalization earlier, to limit outdoor activities and/or habitually wear a facemask when going outside on days with high levels of PM_2.5_ and ozone concentrations. This will help minimize personal exposure to air pollutants. The parents or families with younger children with asthma should also be more cautious about their children’s outdoor activities on days with high levels of ambient air pollutants and particularly during the warm season. Second, the results of this study can help health care professionals, who serve pediatric patients with asthma, to understand that ambient air pollutants, including PM_2.5_ and ozone, could increase the risk of preventable hospital readmission among children with asthma. Third, this study’s findings may help policy makers to develop a policy to control ambient air pollution in order to reduce emissions of PM_2.5_ and ozone.

Fourth, previous evidence showed that pre-discharge patient/family education is a successful intervention to reduce preventable readmission [42,43]. Therefore, it would be important for hospital leaders or health care workers in children’s hospitals to consider emphasizing the contents related to ambient air pollution when delivering education to the children with asthma and their family before discharge as one of the readmission reduction initiatives. The educational contents may include the effects of ambient air pollution, federal standards for each air pollutant, and how to check daily outdoor air quality and prevent air pollutant exposure. Finally, children’s hospitals will be able to collaborate with asthma education/prevention programs in the community to follow up with pediatric patients with asthma more effectively, given a study’s argument that optimal inpatient asthma care includes an effective transition to the community with constant follow-up care to prevent repeated hospitalization [42]. Home-visit asthma education has especially been proven to be effective in improving asthma-related health outcomes for the population living in disadvantaged communities with limited access to health care and education [44,45,46]. The community-hospital partnership for education and follow up for pediatric patients with asthma may play a significant role in reducing preventable hospital readmission [47]. 

## 5. Conclusions

This is the first study, to the best of our knowledge, which investigates the association between short-term exposures to ambient air pollution and preventable hospital readmissions for pediatric asthma patients in South Texas. Our study confirms the adverse effects of PM_2.5_ and ozone concentrations on preventable hospital readmission among children with asthma in low-income communities. We discovered that younger age and exposure during the warmer season were associated with the effects of ambient air pollutants. Our findings contribute to the limited scientific evidence regarding the effect of ambient air pollutants on hospital readmission for pediatric asthma. However, further research is still warranted to confirm our findings. 

## Figures and Tables

**Figure 1 ijerph-17-04846-f001:**
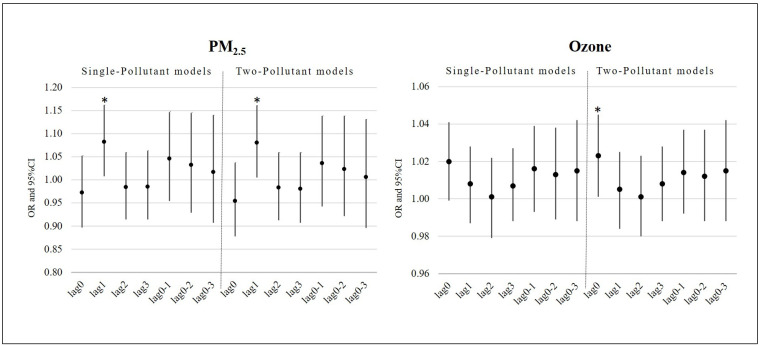
Odds ratio (OR) and 95% confidence interval (CI) of PM_2.5_ and ozone with hospital readmissions for children with asthma in the single- and two-pollutant models (* *p* < 0.05).

**Table 1 ijerph-17-04846-t001:** Descriptive characteristics of asthma pediatric patients readmitted to the hospital between 2010 and 2014 (*N* = 111).

Variable	Mean ± SD [Min, Max] or *N* (%)
**Age (years)**	9.54 ± 3.34 [5, 18]
**Age**	
5–11 years old	88 (79.3)
12–18 years old	23 (20.7)
**Gender**	
Female	47 (42.3)
Male	64 (57.6)
**Ethnicity**	
Hispanic	88 (79.3)
Non-Hispanic	23 (20.7)
**Type of insurance**	
Public (Medicaid)	78 (70.3)
Private	29 (26.1)
Self-pay	4 (3.6)
**Days to readmission (days)**	386.5 ± 364.24 [1, 1651]
1–30 days	9 (8.1)
31–90 days	12 (10.8)
91–180 days	19 (17.1)
181–365 days	30 (27.1)
366 days or longer	41 (36.9)
**Season**	
Warm (May–October)	53 (47.7)
Cold (November–April)	58 (52.3)
**Year**	
2010	13 (11.7)
2011	22 (19.8)
2012	18 (16.2)
2013	26 (23.4)
2014	32 (28.8)

Note: SD—standard deviation.

**Table 2 ijerph-17-04846-t002:** Summary statistics of daily ambient air pollutant concentrations and daily temperature in South Texas between 2010 and 2014.

Variable	Mean	SD	Minimum	Percentile			Maximum	IQR
				25	50	75		
PM_2.5_ (μg/m^3^)	8.30	1.49	4.66	7.19	8.25	9.19	12.37	2.0
Ozone (ppb)	37.37	6.81	20.53	31.47	37.24	42.11	52.84	10.64
Temperature (°C)	20.20	5.85	8.83	16.31	21.01	26.91	29.73	10.6

Note: SD—standard deviation; IQR—interquartile ranges.

**Table 3 ijerph-17-04846-t003:** Results of the conditional logistic regression models between ambient air pollution and hospital readmission.

Single- and Cumulative-Day Lags	Single-Pollutant Model				Two-Pollutant Model			
	PM_2.5_		Ozone		PM_2.5_		Ozone	
	OR (95% CI)	*p*-Value	OR (95% CI)	*p*-Value	OR (95% CI)	*p*-Value	OR (95% CI)	*p*-Value
Lag0	0.972 (0.897–1.052)	0.479	1.020 (0.999–1.041)	0.064	0.954 (0.878–1.037)	0.272	1.023 (1.001–1.045)	0.042 *
Lag1	1.082 (1.008–1.162)	0.030 *	1.008 (0.987–1.028)	0.469	1.080 (1.005–1.161)	0.036 *	1.005 (0.984–1.025)	0.665
Lag2	0.984 (0.914–1.059)	0.664	1.001 (0.979–1.022)	0.955	0.983 (0.912–1.059)	0.655	1.001 (0.980–1.023)	0.904
Lag3	0.985 (0.914–1.063)	0.700	1.007 (0.988–1.027)	0.472	0.980 (0.907–1.059)	0.613	1.008 (0.988–1.028)	0.429
Lag0–1	1.046 (0.954–1.147)	0.334	1.016 (0.993–1.039)	0.168	1.036 (0.942–1.138)	0.468	1.014 (0.992–1.037)	0.222
Lag0–2	1.032 (0.929–1.145)	0.558	1.013 (0.989–1.038)	0.292	1.023 (0.921–1.138)	0.667	1.012 (0.988–1.037)	0.667
Lag0–3	1.017 (0.907–1.140)	0.779	1.015 (0.988–1.042)	0.275	1.006 (0.896–1.131)	0.915	1.015 (0.988–1.042)	0.290

Note: Models adjusted for temperature; * *p* < 0.05.

**Table 4 ijerph-17-04846-t004:** Results of the conditional logistic regression models stratified by age.

Single- and Cumulative-Day Lags	5–11 Years Old				12–18 Years Old			
	Single-Pollutant		Two-Pollutant		Single-Pollutant		Two-Pollutant	
	PM_2.5_	Ozone	PM_2.5_	Ozone	PM_2.5_	Ozone	PM_2.5_	Ozone
Lag0	0.934 (0.850–1.026)	1.022 (0.999–1.047)	0.908 (0.821–1.004)	1.029* (1.004–1.055)	1.123 (0.942–1.337)	1.009 (0.964–1.057)	1.122 (0.942–1.336)	1.008 (0.965–1.054)
Lag1	1.075 (0.994–1.162)	1.009 (0.986–1.033)	1.071 (0.990–1.159)	1.006 (0.982–1.030)	1.112 (0.924–1.338)	1.001 (0.959–1.046)	1.113 (0.924–1.339)	1.002 (0.960–1.045)
Lag2	0.962 (0.881–1.050)	1.003 (0.980–1.027)	0.959 (0.878–1.048)	1.005 (0.981–1.029)	1.045 (0.908–1.204)	0.983 (0.933–1.034)	1.062 (0.917–1.230)	0.978 (0.927–1.031)
Lag3	0.983 (0.905–1.067)	1.004 (0.982–1.026)	0.980 (0.902–1.066)	1.005 (0.983–1.027)	0.969 (0.795–1.182)	1.016 (0.968–1.067)	0.953 (0.779–1.166)	1.019 (0.969–1.071)
Lag0–1	1.017 (0.917–1.127)	1.018 (0.993–1.044)	1.000 (0.899–1.112)	1.018 (0.992–1.045)	1.174 (0.945–1.459)	1.006 (0.958–1.056)	1.176 (0.946–1.462)	1.007 (0.960–1.055)
Lag0–2	0.994 (0.880–1.122)	1.016 (0.989–1.044)	0.982 (0.868–1.111)	1.017 (0.989–1.045)	1.152 (0.923–1.437)	1.000 (0.948–1.056)	1.153 (0.923–1.440)	0.997 (0.946–1.051)
Lag0–3	0.983 (0.861–1.121)	1.016 (0.987–1.047)	0.970 (0.848–1.110)	1.017 (0.987–1.048)	1.123 (0.879–1.435)	1.006 (0.948–1.067)	1.122 (0.877–1.435)	1.003 (0.946–1.063)

Note: Models adjusted for temperature; * *p* < 0.05.

**Table 5 ijerph-17-04846-t005:** Results of the conditional logistic regression analysis stratified by gender.

Single- and Cumulative-Day Lags	Girls				Boys			
	Single-Pollutant		Two-Pollutant		Single-Pollutant		Two-Pollutant	
	PM_2.5_	Ozone	PM_2.5_	Ozone	PM_2.5_	Ozone	PM_2.5_	Ozone
Lag0	0.936 (0.819–1.070)	1.021 (0.989–1.054)	0.902 (0.776–1.048)	1.029 (0.993–1.066)	0.994 (0.900–1.099)	1.019 (0.991–1.047)	0.983 (0.888–1.089)	1.019 (0.991–1.048)
Lag1	1.113 (0.997–1.244)	1.012 (0.984–1.042)	1.108 (0.989–1.240)	1.007 (0.977–1.037)	1.061 (0.965–1.166)	1.002 (0.973–1.032)	1.061 (0.965–1.166)	1.001 (0.972–1.031)
Lag2	0.972 (0.858–1.102)	1.003 (0.973–1.035)	0.967 (0.849–1.101)	1.006 (0.974–1.038)	0.986 (0.900–1.079)	0.996 (0.967–1.027)	0.986 (0.900–1.080)	0.996 (0.967–1.027)
Lag3	0.984 (0.875–1.107)	1.007 (0.979–1.037)	0.972 (0.858–1.100)	1.010 (0.980–1.041)	0.985 (0.892–1.087)	1.005 (0.979–1.033)	0.984 (0.891–1.087)	1.006 (0.979–1.033)
Lag0–1	1.053 (0.911–1.218)	1.018 (0.986–1.052)	1.032 (0.885–1.203)	1.016 (0.983–1.051)	1.044 (0.926–1.175)	1.012 (0.982–1.044)	1.038 (0.921–1.171)	1.011 (0.980–1.043)
Lag0–2	1.038 (0.882–1.222)	1.016 (0.981–1.052)	1.019 (0.859–1.208)	1.015 (0.979–1.052)	1.026 (0.897–1.175)	1.009 (0.974–1.044)	1.025 (0.895–1.173)	1.008 (0.974–1.044)
Lag0–3	1.015 (0.852–1.209)	1.016 (0.979–1.054)	0.989 (0.821–1.193)	1.017 (0.978–1.057)	1.013 (0.872–1.176)	1.011 (0.973–1.050)	1.012 (0.871–1.176)	1.011 (0.973–1.050)

Note: Models adjusted for temperature.

**Table 6 ijerph-17-04846-t006:** Results of the conditional logistic regression analysis stratified by season.

Single- and Cumulative-Day Lags	Warm Season (May–October)				Cold Season (November–April)			
	Single-Pollutant		Two-Pollutant		Single-Pollutant		Two-Pollutant	
	PM_2.5_	Ozone	PM_2.5_	Ozone	PM_2.5_	Ozone	PM_2.5_	Ozone
Lag0	1.056 (0.941–1.184)	1.043 ** (1.012–1.075)	0.993 (0.874–1.127)	1.043 * (1.010–1.078)	0.920 (0.818–1.034)	1.023 (0.986–1.062)	0.922 (0.818–1.038)	1.022 (0.985–1.061)
Lag1	1.134 * (1.019–1.262)	1.016 (0.989–1.043)	1.125 * (1.009–1.254)	1.009 (0.982–1.037)	1.032 (0.933–1.141)	1.013 (0.975–1.053)	1.030 (0.931–1.138)	1.013 (0.975–1.052)
Lag2	1.038 (0.933–1.155)	1.003 (0.976–1.031)	1.037 (0.930–1.158)	1.001 (0.973–1.030)	0.934 (0.837–1.042)	1.008 (0.970–1.047)	0.932 (0.835–1.041)	1.009 (0.970–1.050)
Lag3	1.011 (0.915–1.116)	1.013 (0.989–1.039)	0.995 (0.895–1.106)	1.014 (0.988–1.041)	0.956 (0.853–1.071)	1.011 (0.973–1.051)	0.956 (0.853–1.071)	1.011 (0.972–1.052)
Lag0–1	1.146 * (1.003–1.309)	1.033 * (1.002–1.065)	1.105 (0.961–1.271)	1.025 (0.993–1.059)	0.965 (0.844–1.104)	1.023 (0.982–1.066)	0.968 (0.846–1.108)	1.022 (0.981–1.066)
Lag0–2	1.151 (0.989–1.340)	1.026 (0.994–1.060)	1.123 (0.960–1.314)	1.020 (0.987–1.054)	0.926 (0.793–1.081)	1.023 (0.977–1.071)	0.927 (0.795–1.082)	1.023 (0.977–1.072)
Lag0–3	1.132 (0.965–1.329)	1.031 (0.997–1.066)	1.098 (0.929–1.296)	1.025 (0.990–1.062)	0.892 (0.744–1.070)	1.029 (0.976–1.085)	0.894 (0.747–1.070)	1.029 (0.975–1.087)

Note: Models adjusted for temperature; * *p* < 0.05 ** *p* < 0.01.

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
