# Peer review of "Effect of Ambient Air Pollution on Hospital Readmissions among the Pediatric Asthma Patient Population in South Texas: A Case-Crossover Study"

_ijerph, 2020, doi:10.3390/ijerph17134846_

Round 1
Reviewer 1 Report
The present study investigated the effect of ambient air pollution on hospital readmissions among pediatric asthma patients from South Texas population. The data showed that PM2.5 concentration and in two-pollutant model both PM2.5 and ozone had a significant positive effect on risk for hospital readmissions. The authors also showed that the effect of ambient air pollutants on hospital readmissions was increased in younger children and in warm season.
The research topic is of huge meaning for asthma patients and clinicians, data collection seems all right, the text was well written. The study has some limitation, such as not accurate assessment of the temperature and air pollutant concentration and limited patient population to one hospital, however authors are aware of that and describe the problem in the text.
I recommend to accept the paper for publication in the journal after minor correction. I have one remark:
- In the Introduction lines 68-78 and in the Discussion lines 220-226 authors repeated the same information from the same references. I would suggest to leave them only in the discussion section.
Author Response
Response Letter
Effect of ambient air pollution on hospital readmissions among pediatric asthma patient population in South Texas: A case-crossover study.
Reviewer 1.
The present study investigated the effect of ambient air pollution on hospital readmissions among pediatric asthma patients from South Texas population. The data showed that PM2.5 concentration and in two-pollutant model both PM2.5 and ozone had a significant positive effect on risk for hospital readmissions. The authors also showed that the effect of ambient air pollutants on hospital readmissions was increased in younger children and in warm season.
The research topic is of huge meaning for asthma patients and clinicians, data collection seems all right, the text was well written. The study has some limitation, such as not accurate assessment of the temperature and air pollutant concentration and limited patient population to one hospital, however authors are aware of that and describe the problem in the text.
I recommend to accept the paper for publication in the journal after minor correction. I have one remark:
- In the Introduction lines 68-78 and in the Discussion lines 220-226 authors repeated the same information from the same references. I would suggest to leave them only in the discussion section.
Response: Thank you for your suggestion. We left the information only in the discussion section and deleted the same information in the introduction section (lines 68-81 with track change).

Reviewer 2 Report
Review of ijerph-850284 Effect of ambient air pollution on hospital readmissions among pediatric asthma patient population in South Texas: A case-crossover study.
This is a well-done article that should be published with minor edits, suggested below.
Line 33. This says that exposures were “short-term”, but for ambient air, would such exposures not be long-term? In any case, the duration of exposure might be better defined in the abstract. It is also noteworthy that the odds ratios are only slightly above 1, even if they are statistically significant.
The abstract does not mention if demographics (e.g., income, race) were significant modifiers, as well as access to medical care and insurance.
Were other asthma-related pollutants, notably NO2 also considered in the model?
Line 47. This paragraph notes the importance of indoor pollutants, but the studies referenced are old. Consider referencing a more recent study on asthma and housing quality, such as Takaro The Breathe-Easy Home: The Impact of Asthma-Friendly Home Construction on Clinical Outcomes and Trigger Exposure Am J Public Health. 2011;101:55–62. doi:10.2105/AJPH.2010. 300008. The evidence of home-based exposures on asthma is better reviewed here: Krieger J. Home Is Where the Triggers Are: Increasing Asthma Control by Improving the Home Environment PEDIATRIC ALLERGY, IMMUNOLOGY, AND PULMONOLOGY Volume 23, Number 2, 2010
Line 119. The proximity of the air pollutant monitoring station to the patent’s home might be described in here somewhere. The article only says “South Texas”, which would be a pretty large area.
The limitations to the study are well described.
Line 288. A face mask likely has little effect on ozone inhalation
The implication that PM2.5 and ozone emissions should be reduced to help prevent hospital readmissions is not mentioned.
Author Response
Response Letter
Effect of ambient air pollution on hospital readmissions among pediatric asthma patient population in South Texas: A case-crossover study.
Reviewer 2.
This is a well-done article that should be published with minor edits, suggested below.
Line 33. This says that exposures were “short-term”, but for ambient air, would such exposures not be long-term? In any case, the duration of exposure might be better defined in the abstract. It is also noteworthy that the odds ratios are only slightly above 1, even if they are statistically significant.
Response: Thank you for your point. We included the duration of exposure used in this study in the abstract. Due to the word limits in the abstract (200 words), we added only “(4 days)” after short-term. Also, as you mentioned, the odds ratios in this study were only a little bigger than 1. We assume that this is because the study region did not have high air pollution levels and a large variation in each air pollutant among census tracts. Further studies would be needed in different settings to confirm this association.
The abstract does not mention if demographics (e.g., income, race) were significant modifiers, as well as access to medical care and insurance.
Response: We appreciate your comments. In this study, potential confounders, such as individual-level characteristics (demographics: income, race and access to medical care and insurance), were controlled by the study design (a case-crossover study design) since case and control periods are compared for the same patient. We mentioned this information in the Materials and Methods section (Study Design and Measurement: lines 130-132 with track change) although it was not included in the abstract due to the word count limit. In addition, we only included age, gender, and season in the stratified models given the previous studies. Thus, we do not know if the factors you mentioned are significant modifiers.
Were other asthma-related pollutants, notably NO2 also considered in the model?
Response: Thank you for your question. We were not able to include other asthma-related pollutants like NO2 in this study since the data were not available. We added this information in the limitations section: “Fourth, some important factors that may potentially be associated with personal exposure to air pollution, such as the amount of time spent for outdoor activities and the level of indoor air pollution, and other air pollutants like NO2 were not included in this study due to unavailability of data.” (lines 289-292 with track change).
Line 47. This paragraph notes the importance of indoor pollutants, but the studies referenced are old. Consider referencing a more recent study on asthma and housing quality, such as Takaro The Breathe-Easy Home: The Impact of Asthma-Friendly Home Construction on Clinical Outcomes and Trigger Exposure Am J Public Health. 2011;101:55–62. doi:10.2105/AJPH.2010. 300008. The evidence of home-based exposures on asthma is better reviewed here: Krieger J. Home Is Where the Triggers Are: Increasing Asthma Control by Improving the Home Environment PEDIATRIC ALLERGY, IMMUNOLOGY, AND PULMONOLOGY Volume 23, Number 2, 2010
Response: Thank you for suggesting some recent studies. We changed the references reflecting the studies you mentioned (line 52 with track change).
Line 119. The proximity of the air pollutant monitoring station to the patent’s home might be described in here somewhere.
Response: Thanks for your point. The air pollution data were the daily average predicted concentrations collected from the Centers for Disease Control and Prevention (CDC). The data were not obtained from the air pollutant monitoring station directly. The CDC does not provide the information about proximity. Thus, we were not able to include the information about the proximity for air pollutants. On the other hand, we collected the temperature data measured in the nearest air monitoring station from each patient’s residence. The data came from the Texas Commission on Environmental Quality (TCEQ). The information about the proximity is as follows: mean - 5.9 miles, standard deviation - 9.5 miles, minimum - 0.3 miles, and maximum - 60.9 miles. However, we did not mention this information since temperature is not the main independent variable.
The article only says “South Texas”, which would be a pretty large area.
Response: Thank you for your comment. We used “South Texas” since the hospital serves for children from all South Texas. This study also included participants from South Texas areas although most of them lived in several counties around the city of Corpus Christi where the hospital is located.
The limitations to the study are well described.
Response: Thank you for your comment.
Line 288. A facemask likely has little effect on ozone inhalation
The implication that PM2.5 and ozone emissions should be reduced to help prevent hospital readmissions is not mentioned.
Response: We appreciate your point. As you suggested, we added some information in the implications section: “Third, this study’s findings may help policymakers to develop a policy to control ambient air pollution in order to reduce emissions of PM2.5 and ozone.” (lines 305-307 with track change)